# SCALING CONCEPT WITH TEXT-GUIDED DIFFUSION MODELS

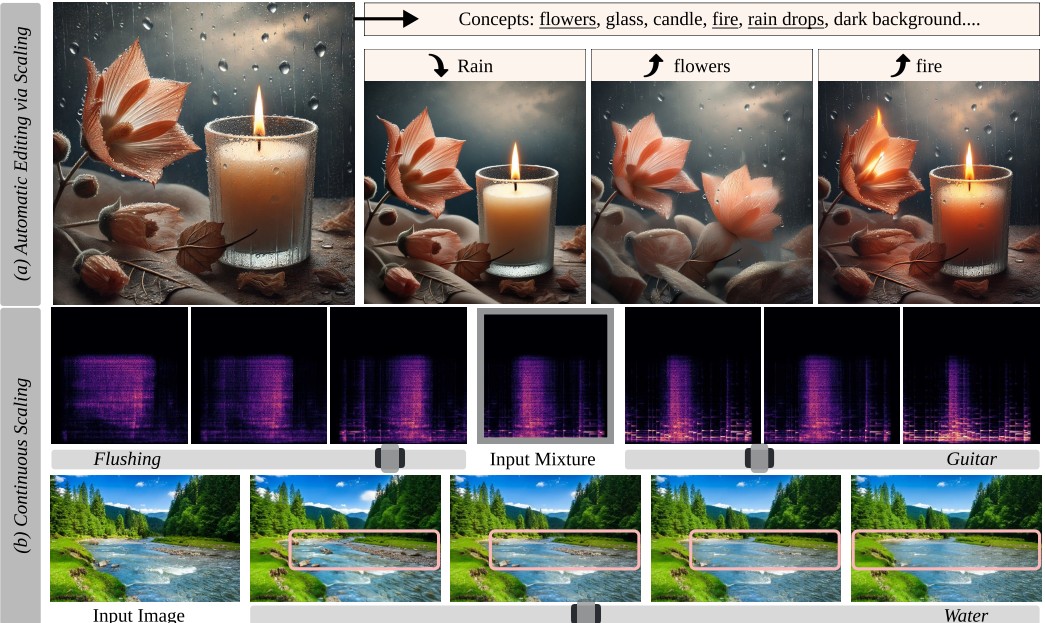

Figure 1: ScalingConcept provides two key functionalities: (a) Automatic Concept Suggestions: It leverages concepts automatically detected in the input, *e.g.*, "fire," "flowers," and "rain" to generate scaling results. This enables automatic editing suggestions, offering users intuitive guidance on potential editing directions. (b) Continuous Concept Scaling: It supports slider-like functionality, allowing users to seamlessly adjust the prominence of a concept across both the audio and image domains.

## ABSTRACT

Text-guided diffusion models have revolutionized generative tasks by producing high-fidelity content based on text descriptions. Additionally, they have enabled an editing paradigm where concepts can be replaced through text conditioning. In this work, we explore a novel paradigm: instead of replacing a concept, can we scale it? We conduct an empirical study to investigate concept decomposition trends in text-guided diffusion models. Leveraging these insights, we propose a simple yet effective method, **ScalingConcept**, designed to enhance or suppress existing concepts in real input without introducing new ones. To systematically evaluate our method, we introduce the *WeakConcept-10* dataset. More importantly, ScalingConcept enables a range of novel zero-shot applications across both image and audio domains, including but not limited to canonical pose generation and generative sound highlighting/removal.

## 1  INTRODUCTION

Derived from non-equilibrium thermodynamics, diffusion models (Sohl-Dickstein et al., 2015) have shown great success in content generation tasks. By defining a Markov chain that gradually injects

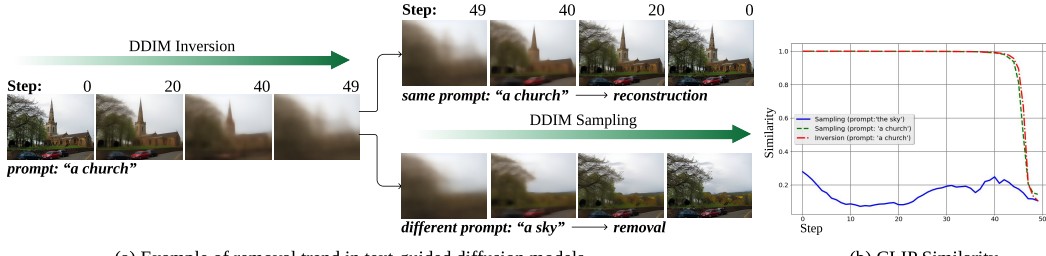

(a) Example of removal trend in text-guided diffusion models      (b) CLIP Similarity

Figure 2: (a) Illustration of concept removal capability observed in the sampling process of text-guided diffusion models when conditioning on a conceptually different prompt compared to the inversion process. (b) We compute the CLIP zero-shot classification results between the classes [*"a sky"*, *"a church"*] and the reconstruction results at each inversion/sampling step (the total number of sampling step is 50), and report the classification accuracy of the class *"a church"*. It's observed that the church object is removed from the removal branch even at the very early stages of sampling.

random noise into data and learning the reverse process, diffusion models generate new content from random noise in an iterative manner. This new generation paradigm has been applied to various domains, such as image generation (Nichol et al., 2022; Ramesh et al., 2022; Saharia et al., 2022; Rombach et al., 2022), video generation (Ho et al., 2022; Singer et al., 2023; Wu et al., 2022; Khachatryan et al., 2023; Guo et al., 2023; Chen et al., 2024; Brooks et al., 2024), and audio generation (Yang et al., 2023; Liu et al., 2023a; Huang et al., 2023b; Ghosal et al., 2023; Liu et al., 2023b; Huang et al., 2023a). Text-guided diffusion models, in particular, have garnered significant attention due to their ability to control content through natural language guidance.

The advent of text-guided diffusion models has enabled text-guided content editing. Several works (Hertz et al., 2023; Gal et al., 2022; Ruiz et al., 2023; Kumari et al., 2023; Brooks et al., 2023; Dhariwal & Nichol, 2021; Song et al., 2020; Mokady et al., 2023) have adapted diffusion models for this purpose. For instance, DreamBooth (Ruiz et al., 2023) fine-tunes a text-to-image diffusion model using a few images of an object paired with a text prompt $c$ that contains the class information of the object. Null-text Inversion (Mokady et al., 2023) addresses the reconstruction error caused by DDIM Inversion (Song et al., 2020) in editing by updating the null-text embedding. LEDITS++ (Brack et al., 2024) improves the accuracy of text-guided editing and supports multiple simultaneous edits. These methods typically focus on addressing a long-standing editing challenge of replacing concepts, such as using an inversion prompt $c$ = *"a dog"* and an editing prompt $c'$ = *"a swimming dog."* While replacement-based paradigms have achieved significant progress in enabling deterministic editing based on clearly defined prompts $c'$, they may fall short in scenarios where users are uncertain about how to specify $c'$. Additionally, certain editing effects are difficult to quantify through text prompts. For example, an instruction such as *"a river with more water"* does not provide an exact specification of the desired increase in water levels, leading to potential ambiguity. Such instructions may correspond to a range of variations in the outcome, as text prompts inherently lack the precision to represent these changes quantitatively.

In this work, we explore a new paradigm beyond the common editing pipeline, which typically involves replacing one concept with another. Instead, we focus on the research question: ***Can we edit the concept continuously without any extra human efforts on specifying a target?*** Specifically, this requires methods capable of isolating concept representations from real input and performing targeted edits on these representations. A surprising finding partially answers this question: text-guided image diffusion models, such as Stable Diffusion (Rombach et al., 2022), exhibit the ability to remove concepts through text prompts. As shown in Figure 2, applying the prompt $c$ = *"a church"* during inversion and the forward prompt $c'$ = *"a sky"* unexpectedly removes the church, while inpainting its region with the neighboring regions. We further investigate this phenomenon by examining its *scalability* and *modality agnosticism*, as detailed in Section 3.2. Through empirical analysis, we observe that the concept removal trend exists on a scalable level, and is not limited to a single modality (both image and audio), proving to be modality-agnostic.

Motivated by the concept removal and reconstruction branches demonstrated in Figure 2, we propose to model the difference between these two branches as a proxy for representing the concept itself,

introducing our method, ScalingConcept. Specifically, given the concept $c$ to be scaled, we apply an inversion technique using text-guided diffusion models to obtain the concept-sensitive latent variable $x_T$. During the sampling process, we model the difference between the noise predictions of the reconstruction and removal branches. A scaling factor is integrated to control the modeling process across different diffusion time steps. Additionally, we introduce a noise regularization term to better balance the fidelity and concept scaling. As shown in Figure 1, our method specializes in the pipeline by modifying the existing concepts in the input, providing editing suggestions without specifying new concepts. Also, by scaling the concept, our method demonstrates a continuous editing capability, such as gradually increasing the water level or making stones disappear progressively. Additionally, our approach interacts solely with the input and output of diffusion models, avoiding intricate modifications to the network's architecture. This design ensures that our approach can be seamlessly applied to diffusion models across various modalities, including audio. Experiments on the public editing dataset TEdBench (Kawar et al., 2023) and our *WeakConcept-10* dataset show that our method outperforms baseline methods in concept scaling, with detailed analysis of the effect of different components.

Interestingly, our zero-shot ScalingConcept method unlocks several downstream applications (as shown in Figure 1) without additional cost. Scaling up a concept standardizes its representation while scaling down tends to remove it. In the image domain, this enables tasks, *e.g.*, canonical pose generation, object stitching, weather manipulation, and creative enhancement. Scaling up adjusts non-standard object poses, completes stitched objects, and harmonizes them with the background. It also allows for altering weather effects, such as deraining or dehazing. In the audio domain, we achieve sound highlighting by amplifying text-indicated sounds and suppressing others, as well as generative sound removal by decomposing audio mixtures into individual components.

In all, our contributions can be summarized as follows:

- We formulate the research question on *concept scaling* and propose ScalingConcept, which has two features: (1) editing the inherent concepts within the input, reducing the effort required for laborious specification of a target, and (2) continuously scaling the concepts along a spectrum, from removal to enhancement.
- To quantitatively validate the effectiveness of ScalingConcept, we introduce a new dataset, *WeakConcept-10*, specifically designed to benchmark concept scaling. We also evaluate its concept suppression capability on the TEdBench (Kawar et al., 2023) dataset. Experimental results demonstrate that our training-free ScalingConcept outperforms baselines across multiple metrics.
- The proposed ScalingConcept showcases its versatility through a variety of zero-shot applications across image and audio domains, such as canonical pose generation, object stitching, weather manipulation, sound highlighting, and generative sound removal, all achieved without additional training. This approach serves as a valuable complement to existing replacement-based editing methods.

## 2 RELATED WORKS

### 2.1 TEXT-GUIDED DIFFUSION MODELS

Text-guided diffusion models have set a new standard for realistic content generation across multiple domains, including images (Nichol et al., 2022; Ramesh et al., 2022; Saharia et al., 2022; Rombach et al., 2022), videos (Ho et al., 2022; Singer et al., 2023; Wu et al., 2022; Khachatryan et al., 2023; Guo et al., 2023; Tang et al., 2024; Brooks et al., 2024), and audio (Yang et al., 2023; Liu et al., 2023a; Huang et al., 2023b; Ghosal et al., 2023; Liu et al., 2023b; Huang et al., 2023a). A major factor contributing to their success is the deep integration of language understanding into the content generation process. For instance, the GLIDE model (Nichol et al., 2022) introduced text-conditional diffusion models that enable controlled image synthesis, while DALL-E 2 (Ramesh et al., 2022) employed a two-stage approach leveraging joint CLIP embeddings (Radford et al., 2021) to capture semantic information from text inputs. Similarly, Imagen (Saharia et al., 2022) showcased the efficacy of large pre-trained language models like T5 (Raffel et al., 2020) in encoding text prompts for image generation tasks. Latent Diffusion Models, such as Stable Diffusion (Rombach et al., 2022), further optimized the diffusion process by performing it in the latent space, enhancing both efficiency and

generation quality. The success observed in the image domain has extended to other modalities. For instance, methods like the Video Diffusion Model (VDM)(Ho et al., 2022), Make-A-Video(Singer et al., 2023), AnimateDiff (Guo et al., 2023), and VideoCrafter (Chen et al., 2023) adapted these models to generate videos from text. In the audio domain, works such as AudioLDM (Liu et al., 2023a), Make-An-Audio (Huang et al., 2023b), and TANGO (Ghosal et al., 2023) have achieved promising results, illustrating the adaptability of diffusion models to various modalities. The success of these models across domains is underpinned by their ability to learn robust text-to-modality associations, proving that textual concepts can be effectively mapped to different types of content. In our work, we build upon these associations, introducing a novel approach to leverage text-guided diffusion models across multiple modalities for the purpose of concept scaling.

## 2.2 Text-guided Editing with Diffusion Models

Text-guided content editing using diffusion models has seen rapid development in recent years. Approaches such as DreamBooth (Ruiz et al., 2023), Null-text Inversion (Mokady et al., 2023), and InstructPix2Pix (Brooks et al., 2023) have introduced techniques to fine-tune and control diffusion models for specific editing tasks. These works focus on replacing or modifying objects within an image by manipulating inversion techniques and null-text embeddings. For instance, DreamBooth (Ruiz et al., 2023) allows for text-guided personalization of diffusion models by fine-tuning them with a small number of images. Null-text Inversion (Mokady et al., 2023) resolves issues related to reconstruction errors when editing specific concepts through prompt-guided inversion. InfEdit (Xu et al., 2023) introduces an inversion-free editing framework that accelerates the editing process while ensuring faithful results. PnP Inversion (Ju et al., 2024) leverages the source diffusion branch to correct inversion deviations, enhancing the accuracy of edits. A recent method LEDITS++ (Brack et al., 2024) provides a novel inversion approach to produce high-fidelity results with a few diffusion steps and supports multiple simultaneous edits. PromptFix (Yu et al., 2024) enhances diffusion models by improving their ability to follow diverse, low-level image editing instructions, while FineMatch (Hua et al., 2024) introduces fine-grained evaluation for text-image alignment, focusing on mismatch detection and correction. In contrast to these methods, which primarily focus on concept replacement, we explore a specific editing paradigm: concept scaling. This approach eliminates the need for explicitly defining instructions, enabling automatic editing suggestions for real-world inputs. Furthermore, it supports continuous editing for scenarios where target instructions are difficult to quantify, offering a more flexible and intuitive editing framework.

## 3 Method

In this section, we first review the foundational concepts of text-guided diffusion models and diffusion inversion techniques in Section 3.1, which form the basis of our analysis. Next, we provide an empirical analysis of the trend of concept decomposition observed in text-guided diffusion models in Section 3.2. Finally, in Section 3.3, we introduce our novel approach, ScalingConcept, which allows flexible control over the strength of the target concept in real input data.

## 3.1 Preliminary

**Text-guided Diffusion Models.** Text-guided diffusion models have gained significant attention for their success in generating realistic images, audio, and video from text prompts. Their key strength lies in accurately capturing text-to-X associations, where X refers to any modality. Taking an image as an example, the process typically begins using an autoencoder such as VQ-GAN (Esser et al., 2021) to project an input into a latent vector $x_0$. During diffusion, Gaussian noise is progressively added to the latent feature, resulting in a random noise vector $x_T$. In the denoising phase, a noise prediction network $\epsilon_\theta$ learns to estimate the noise added at each step. Text-guided diffusion models use a text condition $c$, usually derived from text embeddings like CLIP (Radford et al., 2021), to guide the sequential denoising process. The learning objective is defined as:

$$\ell_{simple} = ||\epsilon - \epsilon_\theta(x_t, c, t)||, \tag{1}$$

where $\epsilon$ is the Gaussian noise added at timestep $t$.

**Inversion Technique.** Inversion techniques are commonly used in generative models to enable the editing of real content (Xia et al., 2022; Gal et al., 2022; Mokady et al., 2023). Typical inversion

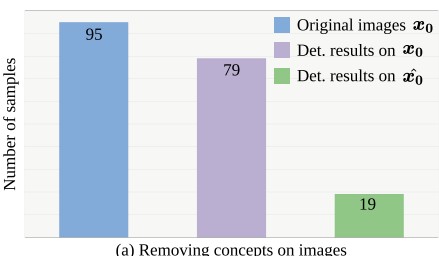 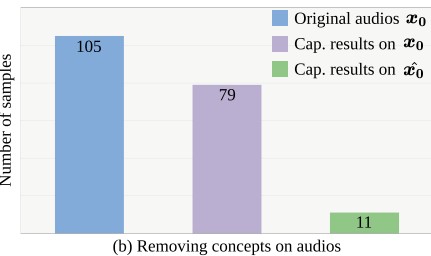

(a) Removing concepts on images     (b) Removing concepts on audios

Figure 3: Analysis of the trend of concept removal. We erase target concepts from given images and audio clips using the proposed inversion and sampling process. We report the number of samples with target concepts before and after concept removal.

methods, such as DDIM inversion (Dhariwal & Nichol, 2021; Song et al., 2020), convert an input latent $x_0$ into a noisy latent variable $x_T$, which can then be used to reconstruct $x_0$ or perform edits. Specifically, DDIM inversion leverages its deterministic sampling process:

$$x_{t-1} = \sqrt{\frac{\bar{\alpha_{t-1}}}{\bar{\alpha}_t}}x_t + \left(\sqrt{\frac{1}{\bar{\alpha_{t-1}}}-1} - \sqrt{\frac{1}{\bar{\alpha}_t}-1}\right)\epsilon_\theta(x_t, c, t), \qquad (2)$$

with $\{\bar{\alpha}_t\}_{t=0}^{T}$ as a predefined noise schedule. This process iteratively denoises $x_T$ to recover $x_0$. Due to ODE formulation, it can be reversed, with small steps, to obtain the inversion (denoted as $f^{inv}(x_t, c, t)$):

$$x_{t+1} = \sqrt{\frac{\bar{\alpha_{t+1}}}{\bar{\alpha}_t}}x_t + \left(\sqrt{\frac{1}{\bar{\alpha_{t+1}}}-1} - \sqrt{\frac{1}{\bar{\alpha}_t}-1}\right)\epsilon_\theta(x_t, c, t), \qquad (3)$$

thereby estimating the noisy latent $x_T$ from $x_0$. Starting with $x_T$, the sampling process can be guided by arbitrary text conditions. However, DDIM inversion is limited by cumulative errors at each step, which deviate the path toward the correct latent noise. Several methods, such as DDPM inversion (Huberman-Spiegelglas et al., 2024) and ReNoise (Garibi et al., 2024), have been proposed to improve the inversion process.

## 3.2 EMPIRICAL ANALYSIS ON THE CONCEPT REMOVAL

Equation (3) and Equation (2) define a pair of destruction and reconstruction processes. In prior research, this framework has been successfully utilized for concept editing. Given an input $x_0$, the inversion process extracts the latent variable $x_T$. The reverse process generates an edited output where the original concept $c$ is modified to $\tilde{c}$, enabling various forms of editing such as object or style changes (e.g., *"a photo of a dog"* → *"a photo of a horse"*). While previous work has focused on replacing the concept with a new one, our research asks a different question: can the existing concept be enhanced or suppressed?

We explore the first question through a case study illustrated in Figure 2. We perform an inversion with the prompt *"a church,"* which branches into two sampling paths: (1) using the same prompt, "a church," to reconstruct the image as expected, and (2) using the prompt *"a sky."* Interestingly, on the second path, the church is removed, and the vacated area is inpainted with content related to the surrounding context, even from the first sampling step. We hypothesize that this removal effect is due to the interplay between cross- and self-attention mechanisms in diffusion models. During inversion, the noise estimator $\epsilon_\theta$ relies heavily on cross-attention to incorporate context from $c$, leading to the strongest modification in regions associated with the concept $c$. However, during sampling, when the prompt *"a sky"* provides no useful context for reconstructing the church, self-attention becomes dominant, leading to the church's removal.

**Does the Concept Removal Trend Appear on Scale?** To determine if the concept removal phenomenon is isolated or consistent across a broader dataset, we replicate the process from Figure 2 using more samples from the COCO (Lin et al., 2014) dataset. For each image $x_0$, we apply the DDIM inversion with the prompt "[class]." After obtaining the noisy latent variable $x_T$, we use a

null prompt $\emptyset$ for the sampling process to convert $\boldsymbol{x_T}$ back into an image $\hat{\boldsymbol{x_0}}$. Note that we use the null prompt for all images as a versatile solution for the removal branch. However, the null prompt can be automatically replaced, as described in Section 3.3. This process mirrors that in Figure 2, aiming to remove the concept of "[class]" from the input image. To assess whether the concept was successfully removed, we used Grounding DINO (Liu et al., 2023c) to detect the presence of the "[class]" object in both $\boldsymbol{x_0}$ and $\hat{\boldsymbol{x_0}}$. The results, shown in Figure 3, indicate that the target concept corresponding to "[class]" is successfully removed in 80% of the images. This confirms that the concept removal capability exists at scale, rather than being limited to a single sample.

**Does the Concept Removal Apply to Other Modality?** To explore this, we conduct a similar experiment with audio. Using the AVE dataset (Tian et al., 2018), an audio event classification dataset containing clips from 28 sound classes, we randomly sampled 5 audio clips from each class. We employ AudioLDM 2 (Liu et al., 2023b) to perform the same process as in the image-based experiment. To determine whether the concept was removed from the original audio clip, we use EnCLAP (Kim et al., 2024), an audio captioning framework, to generate captions for both $\boldsymbol{x_0}$ and $\hat{\boldsymbol{x_0}}$. We then check whether the word "[class]" appeared in the captions. As shown in Figure 3, the same trend of concept removal was observed in audio, despite its fundamentally different nature compared to images.

**Discussions.** From the empirical analysis above, we observe that starting from the same latent variable $\boldsymbol{x_T}$ obtained by inversion, we can define both a reconstruction branch and a removal branch. This implicitly suggests that text-guided diffusion models possess the ability to **extract a concept**. Building on these findings, an important research question emerges: can we control the divergence between these two branches to achieve concept scaling?

### 3.3 OUR METHOD: SCALINGCONCEPT

Motivated by the difference between the removal and reconstruction branches, we propose **ScalingConcept**, a method designed to decompose the concept from real input and scale it up or down, effectively enhancing or suppressing the corresponding representation in the input. Our method consists of the following steps:

**Step 0 (Optional): Concept Parsing.** To facilitate the scaling of embedded concepts in real-world inputs, an optional preliminary step involves parsing concepts from the input (*e.g.*, an image) using off-the-shelf vision-language models. The parsed concepts can then be leveraged to automatically construct the reconstruction and removal branches. In the removal branch, we utilize the null prompt as a baseline example, as described in the subsequent notation. Additionally, Figure 17 provides an analysis of replacing the null prompt with parsed non-$\boldsymbol{c}$ concepts, highlighting its impact on the editing process.

**Step 1: Generating Scaling Startpoint $\boldsymbol{x_T}$.** Given a real input $\boldsymbol{x_0}$ and a concept $\boldsymbol{c}$ to scale, represented by a text prompt such as *"fire hydrant,"* we use a pre-trained text-guided diffusion model $\epsilon_\theta$ to perform sequential inversion functions as described in Equation (3):

$$\boldsymbol{x_T} = f^{inv}(\boldsymbol{x_0}, \boldsymbol{c}, 0) \circ ... \circ f^{inv}(\boldsymbol{x_{T-1}}, \boldsymbol{c}, T-1). \tag{4}$$

In our experiment, we use ReNoise Garibi et al. (2024) as the inversion technique.

**Step 2: Concept Scaling.** Starting from $\boldsymbol{x_T}$, we define two prompts: the first is the text prompt $\boldsymbol{c}$ used during inversion, corresponding to the reconstruction branch, and the second is the null-text prompt $\emptyset$, representing the removal branch. The noise predictions from the two branches are denoted as $\epsilon_t^\emptyset = \epsilon_\theta(\boldsymbol{x_t}, \emptyset, t)$ and $\epsilon_t^r = \epsilon_\theta(\boldsymbol{x_t}, \boldsymbol{c}, t)$, where the superscript $r$ stands for reconstruction. We model the difference between these two branches by capturing the difference in their noise predictions.

$$\hat{\epsilon}_t = \epsilon_t^\emptyset + \omega_t \cdot (\epsilon_t^r - \epsilon_t^\emptyset). \tag{5}$$

We introduce a scaling factor $\omega_t$ to control the magnitude of the difference at each step $t$. Note that when $\omega_t = 1$, Equation (5) degrades to the vanilla reconstruction branch. A value of $\omega_t < 1$ suppresses the concept, while $\omega_t > 1$ enhances it. Intuitively, during the early steps of inference, the model captures coarse-grained details such as global structure and shape, whereas in the final steps, it focuses on refining high-frequency details (Si et al., 2024). To explore the impact of different designs for $\omega_t$, we express it as $\omega_t = \omega_{base} * \beta(t)$, where $\omega_{base}$ controls the overall strength of scaling, and

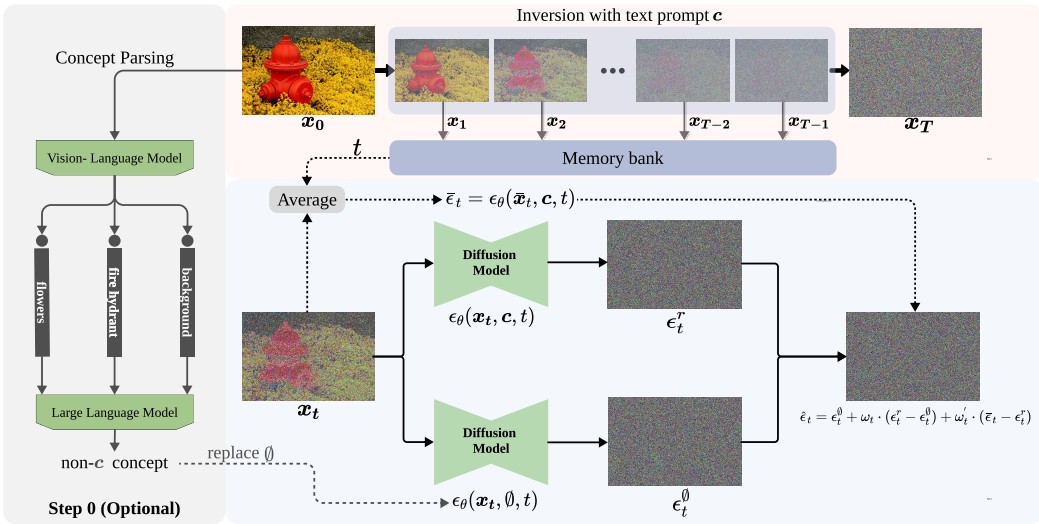

Figure 4: Overview of the ScalingConcept framework. Our method consists of three steps: 0) (Optional) Extracting the embedded concepts from the input by prompting off-the-shelf vision-language models, 1) extracting the latent variable from $x_0$, and 2) constructing different sampling branches and modeling the difference between them.

$\beta(t)$ is a scheduling function within the range 0 to 1. We propose a dynamic schedule $\beta(t) = \left(\frac{t}{T}\right)^{\gamma}$, where $\gamma$ controls the sharpness of the scaling. This approach supports three common schedules: 1) Constant ($\gamma = 0$), treats the difference equally across all steps, similar to classifier-free guidance in diffusion models. 2) Linear ($\gamma = 1$), reflects a linear change in the concept's impact. 3) Non-linear ($\gamma \neq 0$ or 1), allows for dynamic adjustments of the concept's influence, depending on the value of $\gamma$.

**Noise Regularization.** When $\omega_t$ is set to a very large value, the noise prediction $\hat{\epsilon}_t$ in Equation (5) can deviate significantly from the real input, leading to dissimilar content despite the concept being scaled—an undesired effect. Our goal is to scale the concept while preserving the context of the original input. To address this, we introduce a noise regularization term. At each timestep $t$, we retrieve the corresponding noisy latent generated during the inversion process from the memory bank. We combine this with the current noisy latent, adjust the noise predictions using an averaging operation, and then reintroduce them into Equation (6) using the same scaling factor. Additionally, since the forward noisy latents deviate further from the inversion latents in the later steps, we apply an early exit method to stop noise regularization when necessary. The regularized noise prediction is defined as:

$$\hat{\epsilon}_t = \epsilon_t^{\emptyset} + \omega_t \cdot (\epsilon_t^r - \epsilon_t^{\emptyset}) + \omega_t' \cdot (\bar{\epsilon}_t - \epsilon_t^r), \tag{6}$$

$$\omega_t' := \begin{cases} 0 & \text{if } t < t_{exit}, \\ \omega_t & \text{otherwise.} \end{cases} \tag{7}$$

In our experiment, $t_{exit}$ is empirically set to 35, out of a total of 50 sampling steps.

## 4 EXPERIMENT

### 4.1 EXPERIMENTAL SETTINGS

**WeakConcept-10 Dataset.** To effectively test concept scaling, it is essential to have a dataset that supports the measurement of concept strength. However, evaluating whether a concept has been enhanced or suppressed in real inputs poses a significant challenge. To address this, we leverage Stable-Diffusion-3 (SD3) (Esser et al., 2024), a recently released and powerful text-guided image diffusion model, to generate images exhibiting weak concepts. We begin by selecting 10 categories that cover a diverse range of aspects, including *sofa, banana, cat, flower, Van Gogh, ship, Statue*

Table 1: Comparison of different methods for concept enhancement. Results are grouped by dataset: WeakConcept-10 and TEdBench (Kawar et al., 2023). Our method (ScalingConcept) achieves the best performance across multiple metrics.

| Method | WeakConcept-10 | | | TEdBench (Kawar et al., 2023) | | |
|---|---|---|---|---|---|---|
| | FID ↓ | CLIP (%) ↑ | LPIPS ↓ | FID ↓ | CLIP (%) ↓ | SR ↑ (%) |
| *Input* | 313.4 | 26.9 | - | - | 27.3 | - |
| Instruct Pix2Pix | 312.0 | 27.8 | 0.312 | 322.1 | 25.5 | 38.4 |
| LEDITS++ | 274.4 | 28.6 | 0.321 | 316.6 | 22.6 | 58.9 |
| Ours | **272.2** | **28.6** | **0.291** | **315.3** | 22.6 | **69.2** |

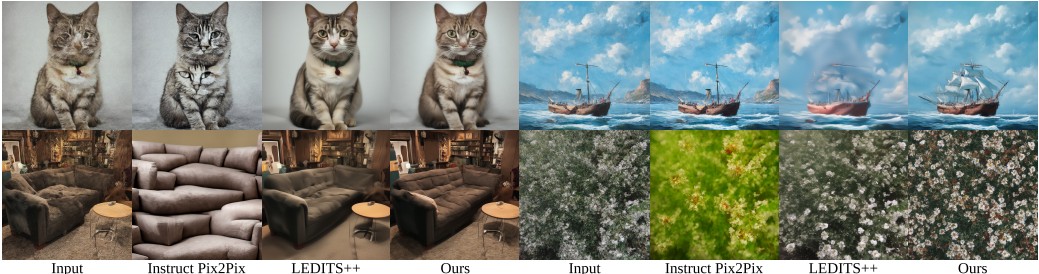

Input    Instruct Pix2Pix    LEDITS++    Ours    Input    Instruct Pix2Pix    LEDITS++    Ours

Figure 5: Qualitative comparison with baseline methods. We display the input images with weak concepts from our dataset, the enhanced results of two baseline approaches, and those of our ScalingConcept method. The concepts being scaled up are "cat," "ship," "sofa," and "flowers", arranged from top-left to bottom-right.

*of Liberty, fruits, forest,* and *horse.* For each category, we generate 10 images using the prompt "[class_name]" while setting the guidance scale to 1, ensuring that the generated images reflect weak representations of the target concept. As illustrated in Figure 18, the generated images display indistinct structures and missing details of the specified concept, making them suitable candidates for improvement through concept scaling. This dataset is particularly for evaluating the concept enhancing (scaling up) performance. We utilize three metrics to evaluate performance: CLIP score (Radford et al., 2021), FID (Heusel et al., 2017), and LPIPS (Zhang et al., 2018). The CLIP score assesses whether the target concept has been successfully enhanced, while FID evaluates the overall image quality after concept enhancement. Finally, LPIPS measures the perceptual similarity between the enhanced output and the original weak input.

**TEdBench (Kawar et al., 2023) Dataset.** We further evaluate the concept scaling-down performance using the public image editing dataset TEdBench (Kawar et al., 2023), which comprises 39 images from diverse categories. For each image, we specify a concept to be scaled down, as detailed in Table 3. To assess performance, we use FID to evaluate the overall image quality after scaling down the concept, CLIP score to measure whether the specified concept has been successfully scaled down, and Success Rate (SR) to quantify the percentage of images where the concept has been successfully scaled down. A common failure mode involves returning the original, unmodified image, which is considered unsuccessful.

## 4.2 MAIN COMPARISON

To evaluate the effectiveness of our ScalingConcept method, we compare it against Instruct Pix2Pix (Brooks et al., 2023), which enhances the concept by using the prompt "enhance the [concept]". Additionally, we adapt another editing method, LEDITS++ (Brack et al., 2024), for our experiment. While LEDITS++ is capable of both adding and removing concepts, in our case, we use it to add the concept again, as the input already contains the concept, effectively simulating concept enhancement. The comparison results are presented in Table 1. Both LEDITS++ and our method achieve comparable concept strength, as indicated by similar CLIP scores. However, our method produces superior image quality, reflected by a lower FID score, while also preserving the original context of the input. This demonstrates the effectiveness of ScalingConcept in both enhancing the concept and maintaining image fidelity. For a qualitative comparison, see Figure 5, where our method clearly enhances the weak concept while preserving fine details in the image. Similarly, we evaluate the scaling-down

Table 2: Ablation studies of our method design. We set $\omega_{base} = 5$ for all experiments. We test the performance with various values of $\gamma$ and examine the impacts of noise regularization and early exit.

| Configuration | Noise Regularization | Early Exit | FID | CLIP (%) | LPIPS |
|---|---|---|---|---|---|
| $\gamma = 0$ (Constant) | ✗ | ✗ | 232.9 | 28.6 | 0.397 |
| $\gamma = 0.5$ (Non-linear) | ✗ | ✗ | 238.6 | 28.7 | 0.380 |
| $\gamma = 1$ (Linear) | ✗ | ✗ | 242.0 | 28.7 | 0.368 |
| $\gamma = 3$ (Non-linear) | ✗ | ✗ | 258.1 | 28.5 | 0.324 |
| $\gamma = 3$ | ✓ | ✗ | 282.6 | 28.5 | 0.260 |
| | ✓ | ✓ | 272.2 | 28.6 | 0.291 |

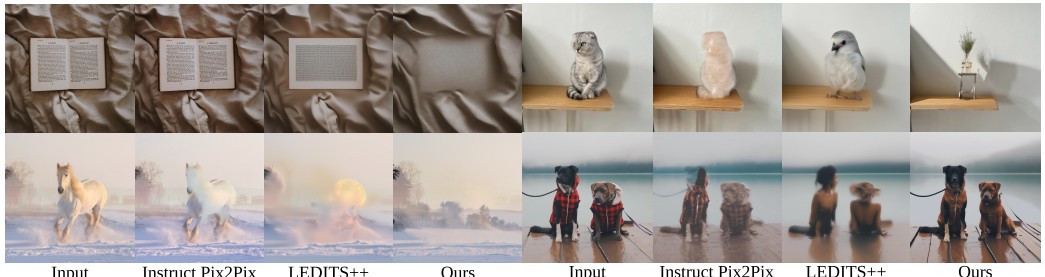

| Input | Instruct Pix2Pix | LEDITS++ | Ours | Input | Instruct Pix2Pix | LEDITS++ | Ours |

Figure 6: Qualitative comparison with baseline methods on concept scaling-down. We present the real input images from the TEdBench dataset alongside the scaling-down results of two baseline approaches and our ScalingConcept method. The concepts being scaled down are "open book," "cat," "horse," and "checkered hoodies", arranged from top-left to bottom-right.

performance, where the goal is to suppress the concept using the TEdBench dataset (Kawar et al., 2023). Our method achieves a lower FID score and a higher success rate (approximately 10% improvement) compared to the strong baseline LEDITS++. Visualization results in Figure 6 further demonstrate that our ScalingConcept method delivers superior concept removal effects. Notably, while LEDITS++ uses a mask to constrain the editing area, this technique can also be incorporated into our method to achieve better region-specific control.

### 4.3 ABLATION STUDIES

In Table 2, we analyze the trade-off between fidelity and generation quality by varying the value of $\gamma$ and introducing noise regularization. We set $\omega_{base} = 5$ for all the ablations. The CLIP score for all variants remains similar (28.5 - 28.7), which demonstrates that $\omega_{base}$ effectively controls the strength of concept scaling. Overall, our goal is to achieve a better balance between concept scaling and content preservation.

**Effect of Different $\gamma$.** As we gradually increase $\gamma$, the FID score rises, indicating that the generated results are shifting from pure generation to a balance between preserving the original content and enhancing the concept (as reflected by the corresponding improvement in the LPIPS score). In this work, we aim to scale the concept, with a focus on achieving a better balance between these factors. Therefore, we select a relatively large value for $\gamma$, such as 3.

**Effect of Noise Regularization and Early Exit.** Introducing the noise regularization term into the method significantly improves the LPIPS score from 0.324 to 0.260, indicating better preservation of the original content. However, this introduces a constraint on concept enhancement. When incorporating early exit, both the FID and CLIP scores improve, while content preservation is slightly compromised, leading to a better overall balance.

### 4.4 ZERO-SHOT APPLICATIONS WITH SCALINGCONCEPT

Our method provides continuous concept scaling up or down for real inputs, making it applicable to a variety of real-world applications. In the audio domain, the continuous scaling capability enables

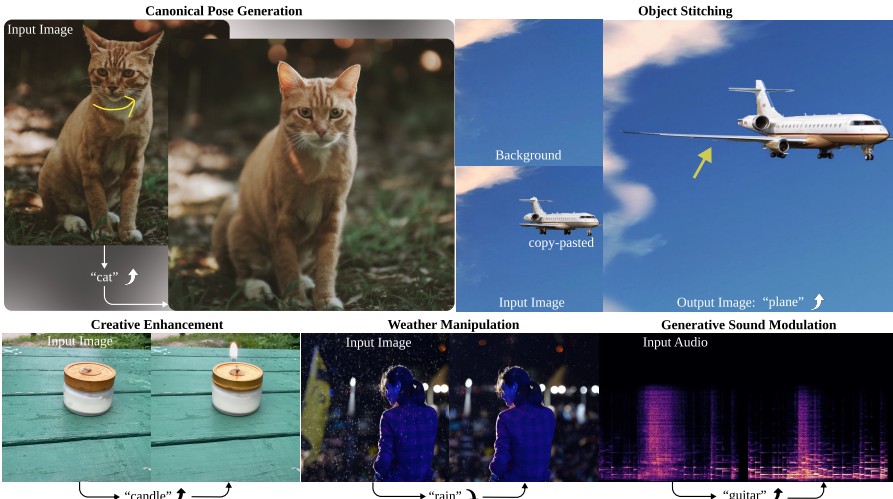

Figure 7: **Applications of ScalingConcept.** We showcase various zero-shot applications across image and audio modalities, highlighting the surprising effects of scaling concepts up or down, including non-trivial tasks like canonical pose generation.

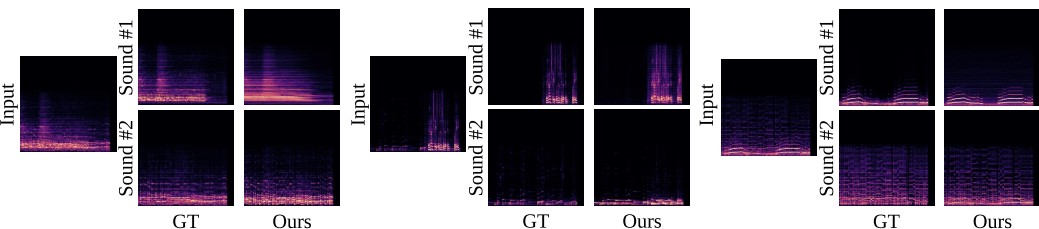

Figure 8: Qualitative comparison on sound separation. Our method enables zero-shot sound removal through a generative model.

sound highlighting, as illustrated in Figure 1. This involves increasing the volume of a target sound by scaling the concept of the corresponding sound category using our approach. Another audio application is sound separation, achieved through a generative model. In Figure 8, we demonstrate this by using a mixture of sounds as input and scaling down the concept of a non-target sound by specifying its class as the inversion prompt. We provide a comparison with the ground truth, showcasing that our method achieves effective sound removal results. In the image domain, our method can also perform a variety of tasks, such as manipulating weather conditions and, intriguingly, adjusting poses, among others. We present a preview of these diverse tasks across different domains in Figure 7. Additional applications can be explored in the Application Zoo, as detailed in Appendix A.1.

## 5 CONCLUSION AND DISCUSSION

We propose ScalingConcept, a zero-shot concept scaling method that focuses on enhancing or suppressing existing concepts in real input data. Our method allows for user-friendly adjustments by freely tuning the scaling strength $\omega_{base}$ and the scaling schedule $\gamma$, to achieve a diverse range of effects. More importantly, ScalingConcept unlocks a variety of non-trivial applications across different modalities, including canonical pose generation and sound removal or highlighting. This approach has the potential to serve as a powerful tool within the growing family of diffusion models. This new method complements existing diffusion-based editing approaches while introducing new challenges, particularly in scaling multiple concepts simultaneously and minimizing unintended effects on other concepts. Existing editing methods have benefited from years of advancements to address similar challenges, such as incorporating attention control. We expect that future work will build on these developments to effectively tackle these challenges for ScalingConcept.

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

## A  APPENDIX

### A.1  APPLICATION ZOO

In this section, we present the application zoo, demonstrating several applications enabled by our ScalingConcept method. Notably, all results are achieved in a zero-shot manner, highlighting the versatility and value of our approach. Additionally, these applications are non-trivial and span across both image and audio domains. For image tasks, we use SDXL Podell et al. (2023) as our base model, while for audio tasks, we employ AudioLDM 2 (Liu et al., 2023b).

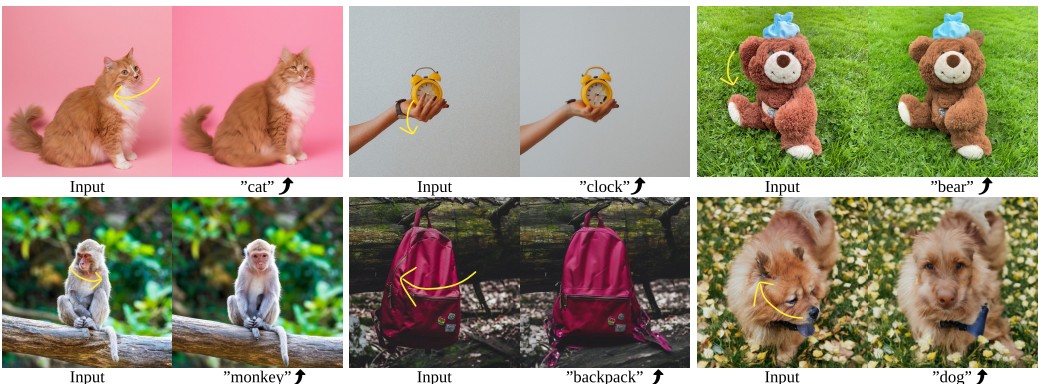

Figure 9: Canonical pose generation. By scaling up the concept of an object, our model adjusts its pose to be more complete and visible.

**Canonical pose generation.** We identify an interesting and non-trivial task enabled by our ScalingConcept method — adjusting the pose of the subject in the image by scaling up the concept. In Figure 9, we demonstrate the canonical pose generation effect. In the original input images, the concepts to be scaled up, such as the cat, clock, and backpack, are depicted in different poses. After applying concept scaling, the cat and backpack are adjusted to face forward, and the clock's occlusion by a hand is mitigated, resulting in a more complete expression of the concept. Across all results, scaling up the concept enables seamless and faithful pose adjustments, a task that is challenging even in the 3D domain, yet is effectively addressed by our method. From a high-level perspective, scaling up the concept strengthens its completeness and visibility, often resulting in front-facing orientations. This technique has potential applications in 3D tasks such as novel-view synthesis.

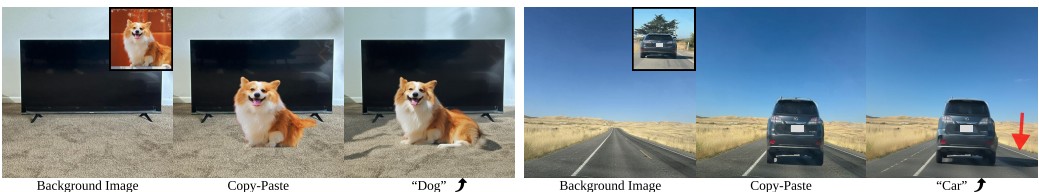

Figure 10: Object stitching. By enhancing an object's concept, we successfully stitch the object and the background together, completing and harmonizing the whole image.

**Object stitching.** Another straightforward application is object stitching. When we copy and paste an object into a background image, we scale up the concept in the copy-paste image, which results in making the object more complete. For example, this can be seen in Figure 10, where the dog is completed, the lighting is adjusted, and the shadow of the car is added.

**Creative Enhancement.** A more open-ended application, as shown in Figure 11, is creative enhancement. In this case, the effect of scaling up the concept is dependent on the actual content of the image, often producing surprising "growing" effects. For example, when scaling up the concept, the *"couple"* transitions from standing separately to holding hands; and the *"pizza"* gains additional toppings. This application is particularly useful when users have an arbitrary image and want to enhance the concept to explore different effects.

**Weather Manipulation.** Since our method supports both scaling up and down concepts, a practical application is weather manipulation (as shown in Figure 12). Scaling down corresponds to classic

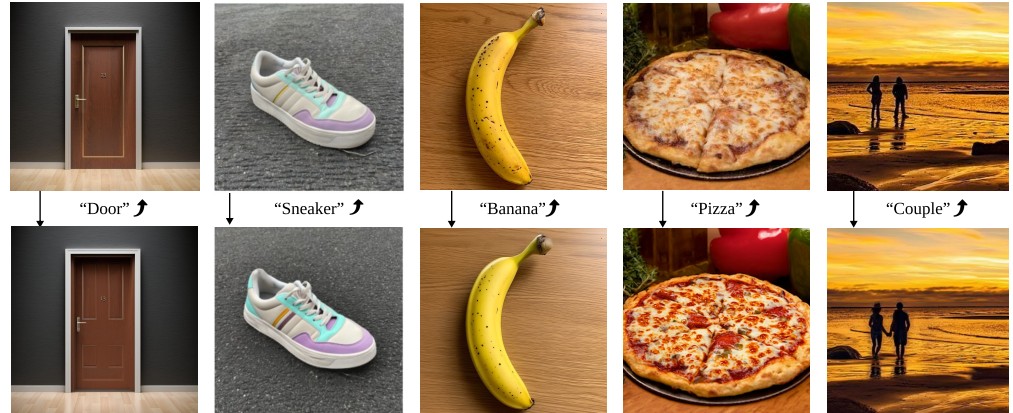

Figure 11: Creative enhancement. ScalingConcept surprisingly produces "growing" effects based on the content of input images.

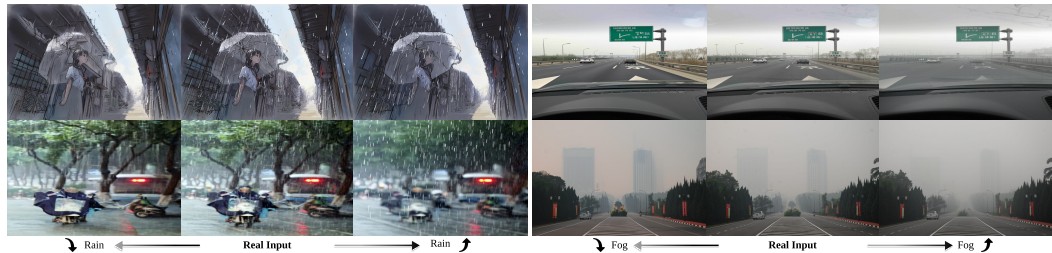

Figure 12: Weather manipulation. Our method enables both weather suppression, similar to deraining and dehazing tasks, and weather enhancement.

weather mitigation tasks, such as deraining or dehazing, while scaling up the weather is useful in scenarios such as movie production, where specific weather conditions are needed. For example, in the movie "The Mist", there is no need to wait for naturally heavy fog—our method can faithfully enhance the fog to achieve the desired effect.

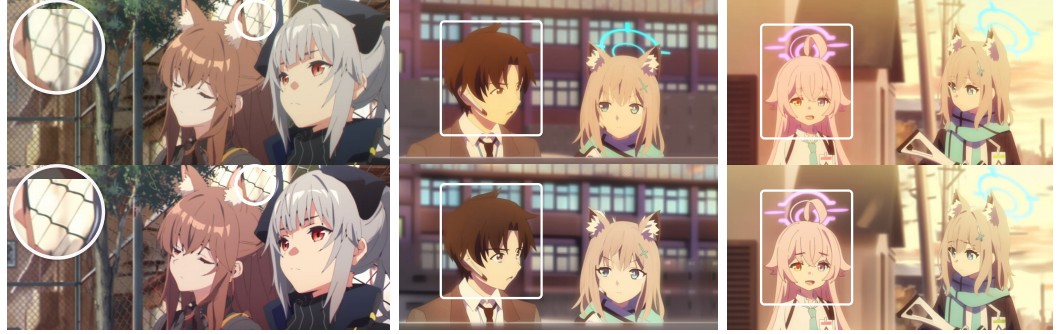

Figure 13: The top row shows screenshots from the anime *"Arknights"* (Left) and *"Blue archive"* (Middle & Right).The bottom row displays the images after scaling up the "anime" concept, which mitigates the fuzziness and blurriness issues commonly encountered in the anime production process.

**Anime Sketch Enhancement.** During the photography and post-production stages of anime making, cumulative errors in line processing often result in blurred lines, making the image appear fuzzy. Filters for scenes like sunsets exacerbate this issue, which cannot be resolved simply by increasing the resolution or bitrate of the anime. Using our ScalingConcept method, we process images with such issues by applying "anime" as the concept to scale up. This enhances the sketches in the image as shown in Fig. 13, leading to an overall improvement in visual clarity.

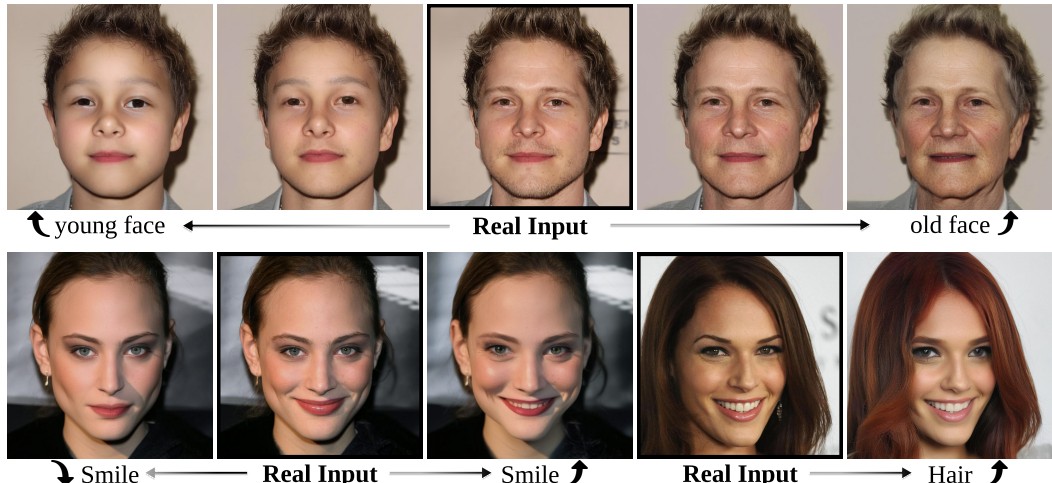

Figure 14: We present a random batch of 3 samples from CelebA-HQ Karras (2017), without cherry-picking, to demonstrate our method's versatility in scaling different face attribute concepts.

**Face Attribute Scaling.** We extend our method to face images. In Figure 14, we showcase popular face attribute editing tasks on examples from the CelebA-HQ Karras (2017) dataset, such as adjusting age, smile, and hair. Each of these edits can be achieved by scaling the corresponding concepts, demonstrating the versatility of our method.

## A.2 IS CANONICAL POSE GENERATION EASY TO ACHIEVE?

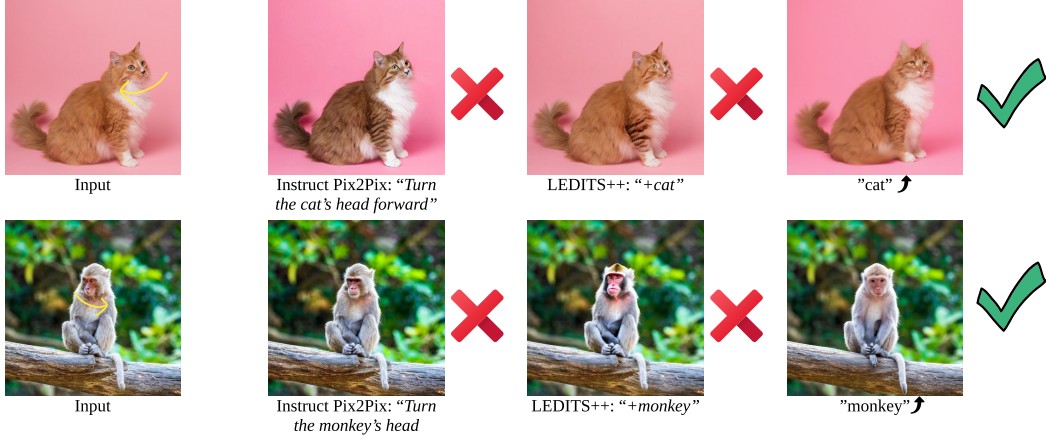

Figure 15: Given the canonical pose generation effect, we attempt to use Instruction Pix2Pix and LEDITS++ to achieve similar results; however, both approaches failed, demonstrating the challenge of this task.

As demonstrated in Fig. 9, our ScalingConcept method can achieve surprising canonical pose generation effects. To further investigate the difficulty of this task, we employ two popular image editing methods: Instruct Pix2Pix Brooks et al. (2023), which follows instructions for editing, and LEDITS++, which adds or removes concepts from the input. Specifically, we instruct Instruct Pix2Pix to "turn the monkey's head forward," but the method fails to produce the desired effect. Similarly, when attempting to add the same concept to the input, LEDITS++ does not achieve the pose generation effect, indicating that this task is non-trivial.

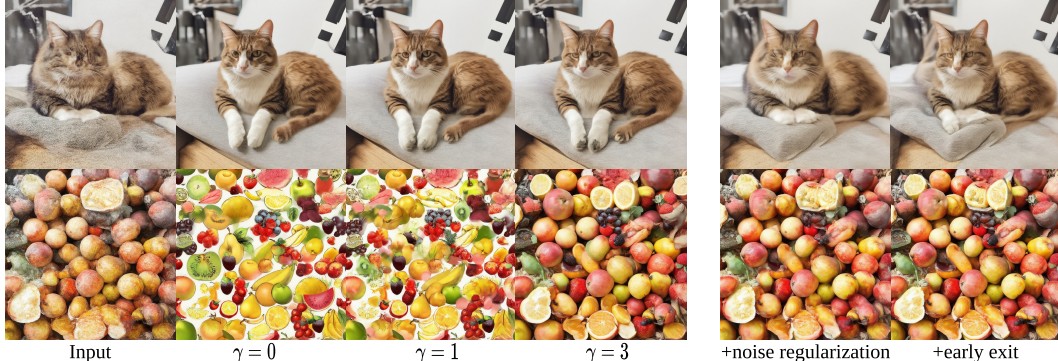

Figure 16: Visualization of ablation studies. We present the results of concept scaling with different method variants.

### A.3 VISUALIZATION OF ABLATION STUDIES

To illustrate the effects of different components of our method, we visualize the results in Fig. 16, which scales up the concepts of "cat" and "fruits" with $\omega_{base} = 5$. The results demonstrate that our non-linear schedule achieves a better trade-off between fidelity and content preservation. Moreover, adding noise regularization helps preserve more fine-grained details, while the introduction of early exit further improves the trade-off.

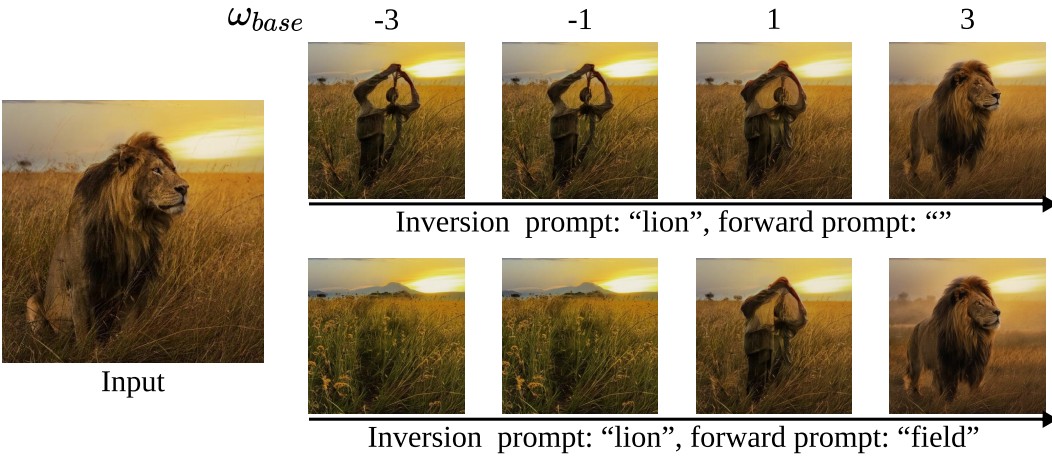

Figure 17: We set $\gamma = 3$ and vary $\omega_{base}$ to investigate its effect. Additionally, we change the prompt from $\emptyset$ to "field" to examine the impact of the forward prompt.

### A.4 EFFECT OF $\omega_{bsae}$

In the previous experiments, we fix $\omega_{base}$ to investigate the effectiveness of other components. In Fig. 17, we showcase the effects of varying $\omega_{base}$, with values ranging from -3 to 3, while fixing $\gamma = 3$. The figure demonstrates that reducing $\omega_{base}$ corresponds to the removal of the concept, whereas increasing it enhances the concept. However, we found that the removal effect is not as satisfactory as the enhancement, which highlights a limitation related to text-to-image association.

### A.5 DOES FORWARD PROMPT MATTER?

In Fig. 17, changing the forward prompt from $\emptyset$ to "field," another concept present in the original input, improves the removal effect, as the region left by the null prompt is inpainted with the concept of "field." This demonstrates the importance of selecting the correct concept to serve as the removal

helper. However, this approach requires additional effort to label the concepts instead of simply using the versatile null prompt. This suggests an advanced setting for the method, where providing coarse-level annotations for an additional concept can lead to significant improvements.

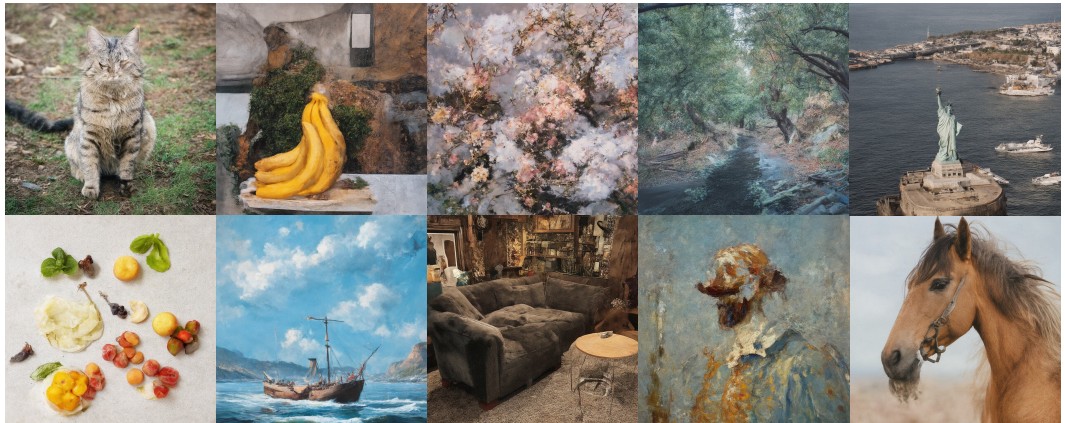

Figure 18: Overview of the WeakConcept-10 dataset. The images exhibit weak and incomplete representations of the target concepts, making them ideal candidates for testing concept scaling methods.

### A.6 DATASET DETAILS

We provide a visualization of the images in our generated WeakConcept-10 dataset in Figure 18. These generated images exhibit indistinct structures and missing details of the specified concepts, making them ideal candidates for improvement through concept scaling.

For experiments involving concept scaling down on TEdBench, we select one concept from each image as the scaling-down candidate. The mapping of images to their corresponding concepts is detailed in Table 3, covering a diverse range of concepts.

### A.7 LIMITATIONS AND FUTURE WORKS

Despite our method presenting a zero-shot approach to scaling concepts in real inputs and achieving promising results, there are several limitations to the current method.

**Choice of Hyperparameters.** In our current method, we split the scaling factor $\omega_t$ into two controlling factors: $\omega_{base}$ and the schedule $\beta(t) = \left(\frac{t}{T}\right)^{\gamma}$. Users can adjust $\omega_{base}$ and $\gamma$ to control the scaling strength. Although we demonstrate the effects of different components in Table 2, the optimal combination varies depending on the task, making user input non-trivial. To address this, a potential future direction is to design an automatic scaling factor that adapts to the target concept's strength, thus eliminating the need for extensive hyperparameter tuning.

**Dependence on Text-to-X Association.** While our method enables concept scaling with text-guided diffusion models for any modality (X), its effectiveness relies heavily on the text-to-X association. If the text prompt is not sensitive to the diffusion model – meaning the information about the concept is not captured effectively – the method may fail. To address this issue, incorporating concept-specific fine-tuning may be beneficial for certain edge cases.

| Image File | Concept to be Scaled Down |
|---|---|
| teddy_1.jpeg | teddy bear |
| flamingo.jpeg | beach |
| dog_with_shirt.jpg | shirt |
| cake_1.jpeg | chocolate shavings |
| chibi.jpeg | cat |
| zebra.jpeg | stripes |
| cat_3.jpeg | cat |
| empty_street.jpeg | Concrete barriers |
| couple_beach.jpeg | couple |
| bird.jpeg | wood |
| dog_01.jpeg | sand |
| white_horse2.png | horse |
| road1.png | road |
| new_cat_3.jpeg | Long fur |
| bird-g83440b9c4_1920.jpg | rope |
| black_shirt.jpeg | watch |
| milk_cookie.jpeg | milk |
| door.jpeg | door |
| giraffe.jpeg | giraffe |
| goat_and_cat.jpg | cat |
| elephant.jpeg | elephant |
| bear3.jpeg | bear |
| two_dogs_with_checkered_shirts1.jpg | checkered hoodies |
| drinking_horse.png | horse |
| tennis_ball.jpeg | ball |
| bird.png | beak |
| egg_tree.jpeg | Nest |
| prague.png | building |
| banana_1.jpeg | banana |
| dog2_standing.png | Green grass |
| chair_1.jpeg | chair |
| box.jpeg | knifes |
| tree_1.jpeg | tree |
| cat.jpeg | cat |
| vase_01.jpeg | flowers |
| apples.jpeg | apples |
| open_book.jpeg | book |
| white_horse1.png | horse |
| red_car.jpeg | Black top |
| pizza1.png | Red pepper |

Table 3: Mapping between image files and the concepts to be scaled down.

