# OpenReview forum: "Scaling Concept With Text-Guided Diffusion Models"
_ICLR.cc/2025/Conference — Submitted to ICLR 2025_

### Official Review · Reviewer_MVQH · 2024-10-25

**Soundness:** 2
**Presentation:** 2
**Contribution:** 2
**Rating:** 5
**Confidence:** 3

**Summary:**

This work explores scaling concepts up or down when performing text-guided image editing using diffusion models. The authors conduct preliminary studies and observe that diffusion models can remove concepts with a null prompt, and this ability can extend to the audio modality. Consequently, they devise a method that combines the reconstruction noise prediction process with the original text prompt and the removal noise prediction process with a null-text prompt. Additionally, they introduce several combination mechanisms, such as constant, linear, and non-linear strategies. Experiments were conducted on a newly constructed dataset, WeakConcept-10, to validate the effectiveness of the proposed method.

**Strengths:**

1. The authors successfully demonstrate that diffusion-based methods can scale concepts across both image and audio modalities.
2. They introduce the WeakConcept-10 dataset.
3. Experiments cover multiple applications in image and audio domains.

**Weaknesses:**

1. The motivation is unclear. The authors' claim that existing editing methods focus primarily on replacing one concept with another seems overly arbitrary and somewhat misleading. Many current editing paradigms modify concepts without direct replacement, such as [1-2].

[1] Direct Inversion: Boosting Diffusion-Based Editing with 3 Lines of Code
[2] Inversion-Free Image Editing with Natural Language

2. This work's main focus is on the inversion process and how to combine the information-preserving and information-modifying branches. However, it appears to overlook relevant prior work in this area.

3. Regarding the hypothesis: The authors hypothesize that the removal effect is due to the interaction between cross- and self-attention mechanisms in diffusion models, but no experiments are provided to support this claim.

4. In the preliminary study, there is a divergence between the observations and the final proposed method. Concept decomposition intuitively suggests the ability to extract different concepts from the input image separately. Yet, the authors only test the ability to remove a concept using a null prompt, without directly demonstrating the decomposition capability. Furthermore, the link between the concept removal ability and the later concept enhancement method is unclear.

5. Is the method limited to enhancing or suppressing existing concepts? If it cannot add or modify new concepts, this work faces significant limitations. Moreover, many current text-editing methods are already capable of object enhancement, such as creative enhancement, weather manipulation, etc.

6. Most experiments focus on concept enhancement without sufficiently validating the model's performance in concept suppression.

7. Formatting issue: In Section A.4, there is an incomplete reference ("in ??").

**Questions:**

1. In the preliminary study, is the prompt “[class].” randomly selected from any class present in the image, or is it only partial? Since the conclusion states that diffusion models can decompose a concept, it's important that this concept can be an arbitrary one.

2. What are the differences and connections between this work and existing research on image enhancement?

[1] Diffusion Models for Image Restoration and Enhancement – A Comprehensive Survey
[2] Generative Diffusion Prior for Unified Image Restoration and Enhancement

---

> ### Author Response · Authors · 2024-11-30
>
> ### Clarification on Motivation
> We appreciate the reviewer’s comments and would like to clarify that our motivation centers on **scaling inherent concepts within real input effectively**. Existing editing methods, such as **InstructPix2Pix** and **AUDIT** [a], focus on providing editing effects through explicit instructions or attention control (e.g., **Prompt-to-Prompt**).
>
> In contrast, our work emphasizes continuous scaling of inherent concepts directly from the input (as shown in Fig. 1). The key challenge lies in obtaining scalable concept representations, inspired by our observations in Sec. 3.2. We have included additional discussion on related works [1,2] as suggested by the reviewer and updated the paper accordingly (L181–184).
>
> [a] AUDIT: Audio Editing by Following Instructions with Latent Diffusion Models
>
> ---
>
> ### Experiment on Cross-Attention Mechanisms
> To further investigate our hypothesis, we follow the approach outlined in [a] to visualize the cross-attention map of an example image (Fig. 2). We then assess the zero-shot probing accuracy using the CLIP model. The results are shown below:
>
> |Class | Church -> Church (Reconstruction) | Church -> Sky (Removal) |
> |-------------|---------------|--------------|
> | ‘Church’     |    0.51     |     0.40    |
> |  ‘Sky’     |    0.49     |     0.60    |
>
> In the removal branch, the cross-attention map is concentrated on the sky region rather than the church. This focused attention contributes to the successful removal of the church concept and the subsequent inpainting with sky elements.
>
> [a] Towards Understanding Cross and Self-Attention in Stable Diffusion for Text-Guided Image Editing
>
> ---
> ### Connection Between Observations and Proposed Method
> As evidenced by common inversion techniques [a,b,c], inversion can reconstruct the concept. In our empirical study, we investigate the ability to remove a concept. Since concept removal and reconstruction begin from the same starting point (see Fig. 2), we define the divergence between these two processes as **the trend of concept decomposition.**
>
> While this does not directly equal concept decomposition, it is pivotal for our method. We model the difference between the removal and reconstruction branches to extract concept-related representations and enable scaling.
>
> [a] Null-text Inversion for Editing Real Images using Guided Diffusion Models
>
> [b] ReNoise: Real Image Inversion Through Iterative Noising
>
> [c] Direct Inversion: Boosting Diffusion-Based Editing with 3 Lines of Code
>
> ---
>
> ### Is the Method Limited to Concept Scaling?
>
> We clarify that our method is tailored for concept scaling but **not limited to this application.** The design of "Step 2: Concept Scaling" (L311–367) focuses on enhancing or suppressing concepts. Disabling this step converts our method into a variant supporting concept addition or modification, similar to ReNoise. Moreover, we want to highlight that beyond concept scaling, our method addresses tasks such as Object Stitching and Audio Removal. These applications are beyond the scope of current image enhancement methods.
>
> ---
>
> ### Experiment on Concept Suppression
> In the revised version, we include additional experiments on **concept suppression** using the **TEdBench** dataset (see Table 1, L412–419). The dataset includes diverse images spanning multiple categories and complex concepts. We compare our method’s performance with existing techniques (L442–462). Our method demonstrates generalization to diverse datasets and surpasses existing methods on TEdBench, showcasing its superior performance.
>
> ---
>
> ### Prompt [class] in Empirical Study
> For the empirical study, the prompt *[class]* is diverse and depends on the dataset:
>
> 1. We use 95 samples from 10 common classes in the COCO dataset. The *[class]* prompt corresponds to the class label for each image.
> 2. We use the AVE dataset which contains 28 sound classes. 5 audio clips are sampled per class, with the *[class]* prompt being the class label.
>
> Additionally, Fig. 1 demonstrates that the choice of concept for a single input image can be arbitrary, supporting the versatility of our approach.
>
> ---
>
> ### Connections to Image Enhancement
> Existing image enhancement methods [1,2] leverage generative priors to address common image corruptions (e.g., low resolution, blur, poor lighting). In contrast, our method focuses on scaling-based editing.
> - **Connections**: Our method can treat certain corruptions as concepts, enabling tasks such as Deraining/Dehazing, Creative Enhancement (Fig. 11), Anime Sketch Enhancement (Fig. 12, 13)
> - **Differences**: Unlike image enhancement methods, our approach extends to tasks such as Canonical Pose Generation (Fig. 9), Object Stitching (Fig. 10), Face Attribute Scaling (Fig. 14), and Audio Removal and Highlighting (Figs. 1, 7, 8)
>
> We will include this discussion in the related work section to clarify the differences and connections between our method and image enhancement techniques.

---

### Official Review · Reviewer_L9w9 · 2024-10-27

**Soundness:** 2
**Presentation:** 3
**Contribution:** 2
**Rating:** 5
**Confidence:** 4

**Summary:**

While traditional methods focus on replacing concepts with new ones, the paper introduces an approach to scale existing concepts. By analyzing how these models decompose concepts, the authors propose a method called ScalingConcept, aiming to either enhance or suppress concepts present in the input without introducing new elements.

**Strengths:**

1.	The paper explores a novel aspect of scaling rather than replacing concepts, which adds a fresh perspective to the application of diffusion models.

2.	ScalingConcept is simple and effective, which indicates practical applicability in enhancing or suppressing concepts.

3.	ScalingConcept's ability to support novel zero-shot applications is across both image and audio domains.

**Weaknesses:**

1.	There is no mention of how ScalingConcept compares to existing methods or techniques, which might raise questions on its relative performance improvements.

2.	The paper does not discuss the complexity of applying ScalingConcept in diffusion models.

3.	It is necessary to demonstrate the method's scalability and effectiveness across diverse datasets beyond WeakConcept-10.

4.	In the application zoo, only a few cases are provided. It is unclear whether the method is effective universally.

**Questions:**

See weakness.

---

> ### Author Response · Authors · 2024-11-30
>
> ### Comparison to Existing Methods or Techniques
> In the experimental section, we provide both quantitative and qualitative comparisons with existing methods, including **InstructPix2Pix** and **LEDITS++**:
>
> 1. **Quantitative Comparison**:
> Table 1 showcases comparisons on **Concept Scaling Up** using our proposed **WeakConcept-10** dataset and **Concept Scaling Down** using the public **TEdBench** dataset. Our method surpasses both InstructPix2Pix and LEDITS++, demonstrating its effectiveness quantitatively.
>
> 2. **Qualitative Comparison**:
> **Figures 5 and 6** include visualizations comparing scaling results with other methods. Our method achieves better scaling results, as evidenced by higher visual fidelity and better alignment with the scaling target.
>
> In all, Section 4.2 provides an in-depth discussion of these comparisons.
>
> ---
>
> ### Complexity of the ScalingConcept Method
> Our ScalingConcept method is simple in implementation and highly adaptable to most text-guided diffusion models:
>
> 1. **Inversion Process**: Most inversion techniques can be used, including the standard **DDIM** or the more advanced **ReNoise**, which is versatile across almost all diffusion models.
>
> 2. **Sampling Process**: Our method only interacts with the **input ($x_t$)** and **output noise predictions ($\epsilon$)** of the diffusion model. It does not modify the internal network architecture, ensuring compatibility with a variety of text-guided diffusion models across different domains (e.g., audio).
>
> ---
>
> ### Scalability and Effectiveness Across Diverse Datasets
>
> We thank the reviewer for their insightful question regarding our method's generalization. In the revised version, we include additional experiments using the **TEdBench** dataset (see Table 1, L412–419), which features diverse images spanning multiple categories and complex concepts.
>
> **Evaluation on TEdBench**: We evaluate concept scaling down on this real-world dataset and compare the performance to existing methods (L442–462). Our method demonstrates strong generalization to diverse inputs and complex concepts, validating its overall effectiveness. Our approach surpasses existing methods on TEdBench, highlighting its robustness and superior performance.
>
> ---
>
> ### More Cases in the Application Zoo
> While the quantitative results in **Table 1** validate the effectiveness of our method, the **Application Zoo** illustrates the versatility of ScalingConcept across diverse downstream tasks. In the revised paper, we include two new tasks to demonstrate the universal applicability of our method (see L805–833, Figs. 13 and 14):
>
> - **Face Attribute Editing**: Adjusting attributes like age and smile.
>
> - **Anime Sketch Enhancement**: Improving and refining anime sketches.
>
> These cases emphasize the practicality and adaptability of our method to both common and novel tasks, further demonstrating its utility in real-world scenarios.
>
> We hope these additions and clarifications address the reviewer’s concerns and demonstrate the broad applicability and simplicity of our method. Thank you for your valuable feedback!

---

### Official Review · Reviewer_VfeD · 2024-10-28

**Soundness:** 2
**Presentation:** 2
**Contribution:** 2
**Rating:** 5
**Confidence:** 5

**Summary:**

This paper proposes a new paradigm for text-to-image generation through the scaling concept. Precisely, the authors control the intensity of the original concept by adjusting the noise difference across two distinct branches (reconstruction and removal): $\hat{\varepsilon}_t = \varepsilon_t^{\varnothing} + \omega_t \cdot \left( \varepsilon_t^r - \varepsilon_t^{\varnothing} \right)$. The authors conducted an empirical study investigating concept decomposition phenomena in current text-guided diffusion models and baseline comparison experiments on the proposed WeakConcept-10 dataset. Additionally, they explore the practical application of the proposed method in various scenarios, like Canonical Pose Generation, Object Stitching, and others.

**Strengths:**

1. The paper is logically structured, and all figures and tables are well-presented.
2. The paper presents a novel motivation, introducing a simple and understandable paradigm for text-to-image generation based on the scaling concept. The authors also conduct empirical investigations on existing image and audio generation tasks to support their claims.
3. The proposed ScalingConcept method seems promising in various downstream tasks, including Object Stitching, Pose Generation, and Creative Enhancement.

**Weaknesses:**

1. The scientific question in this paper is not straightforward. The author seems inspired by their observations without clearly pointing out the flaws of existing methods.
2. The empirical study is inconsistent with the proposed method's idea. For example, Figure 2 shows the removal concepts using different prompts during the reconstruction. However, the proposed method removes image concepts by using a blank prompt.
3. The analysis of Figure 3 lacks illustrative examples. As noted in point 2, replacing a prompt and using a blank prompt are distinct. The former relies more on model capability and a suitable prompt.
4. The main issue with this paper is the unreliable experimental comparison. The authors only compared two baseline methods on the closed dataset WeakConcept-10. Moreover, they did not even analyze the quality of the proposed WeakConcept-10. Therefore, I doubt whether such a comparison is meaningful and reliable.
5. I think amplifying a certain concept in an image inevitably leads to weakening other concepts. For instance, in Figure 7, when the concept of "clock" is emphasized, the "finger" appears distorted. However, the authors do not analyze these shortcomings.
6. Some formatting issues. The references in Appendix A.2 are not displayed correctly.

**Questions:**

1. Why does LPIPS performance degrade when using the Noise Regularization early exit strategy? The author should conduct more analysis on it。
2. The authors should conduct more theoretical analysis rather than an empirical study on the rationality of the proposed method.

---

> ### Author Response · Authors · 2024-11-30
>
> ### Scientific Question in the Paper
> We clarify that our scientific question focuses on **how to scale the inherent concepts within real input effectively**. Existing editing methods in both image and audio domains often emphasize providing editing effects through explicit instructions (e.g., InstructPix2Pix [1] and AUDIT [2]). These methods typically require specialized datasets and fine-tuning of diffusion models or attention control (e.g., [3]).
>
> In contrast, our work emphasizes **continuous scaling of inherent concepts** in the input (as shown in Fig. 1). The key challenge lies in obtaining scalable concept representations directly from the real input, inspired by our observations in Sec. 3.2.
>
> References:
>
> [1] InstructPix2Pix: Learning to Follow Image Editing Instructions. (CVPR 2023)
>
> [2] AUDIT: Audio Editing by Following Instructions with Latent Diffusion Models. (NeurIPS 2023)
>
> [3] Prompt-to-Prompt Image Editing with Cross Attention Control. (ICLR 2023)
>
> ---
>
> ### Usage of Null-Prompt in Empirical Study and Analysis
> We use a null prompt in the empirical study because of its **versatility** across all input types. Our method consists of two branches:
>
> 1. **Reconstruction Branch**: The concept used here is specified by the user during the inversion process.
>
> 2. **Removal Branch**: A null-prompt is employed to avoid additional user effort in selecting another concept.
>
> To illustrate, Fig. 17 compares results using a null-prompt versus a replacement prompt. While the null-prompt achieves the desired removal effect, adding a replacement prompt can enhance results further.
>
> We also propose an automatic and optional step (as shown in Fig.4) involving visual language models (for images) or audio-language models (for audio) to parse the concept and suggest a replacement prompt. This addition enriches the editing space and provides better controllability for users.
>
> ---
>
> ### Experimental Comparison
> We appreciate the reviewer’s feedback regarding the evaluation of our method. In the revised version, we include additional experiments using the widely-used **TEdBench** dataset (see Table 1, L412–419). TEdBench features diverse images with complex concepts, providing a real-world test for our method.
>
> On TEdBench, we evaluate concept scaling down and compare against existing methods (L442–462). The results demonstrate that our method generalizes effectively to diverse datasets and complex concepts, outperforming existing methods.
>
> Additionally, the quality of our proposed **WeakConcept-10** dataset is visualized in Fig. 18. These images exhibit weak and incomplete representations of target concepts, making them ideal for evaluating concept scaling methods.
>
> ---
>
> ### Analysis of Method Shortcomings
> We acknowledge the reviewer’s concern that amplifying a certain concept in an image may weaken other concepts (as shown in the reviewer’s example). This issue could be mitigated by incorporating attention masks to constrain the scaling area, thus reducing interference with other concepts. Attention control is a well-explored area in editing methods ([1, 2, 3]), and we believe such techniques can be integrated into our framework to further enhance its effectiveness in future work (see L535–539).
>
> References:
>
> [1] Prompt-to-Prompt Image Editing with Cross Attention Control (ICLR 2023)
>
> [2] LEDITS++: Limitless Image Editing using Text-to-Image Models (CVPR 2024)
>
> [3] Focus on Your Instruction: Fine-grained and Multi-instruction Image Editing by Attention Modulation (CVPR 2024)
>
> ---
>
> ### LPIPS Performance Degradation with Noise Regularization
> Better LPIPS performance indicates higher perceptual similarity between the scaled result and the original input. Using Noise Regularization without early exit introduces latent variables from inversion at every step, effectively providing the original input during sampling. This unsurprisingly improves LPIPS performance but significantly degrades generation quality (FID increases from 272.2 to 282.6).
>
> To balance image quality and perceptual similarity, we introduce an early exit strategy, which maintains reasonable LPIPS while avoiding a substantial drop in FID.
>
> ---
>
> ### Rationality of the Proposed Method
> Our method is grounded in empirical observations (e.g., concept reconstruction and removal in Fig. 2) and validated across domains (e.g., audio in Sec. 3.2). **The experiments on various datasets (Table1, Fig. 5/6)** for both concept scaling up and scaling down demonstrate the method’s effectiveness compared to existing editing approaches.
>
> Additionally, the **Application Zoo** highlights diverse application scenarios (see Fig. 7/8 and Appendix A.1), further showcasing the generalizability and practicality of our method.
>
> While we believe the empirical evidence supports the method’s validity, we acknowledge that theoretical analysis could strengthen our work. This is an area we plan to explore in future research.
>
> ---
>
> ### Formatting Issue
> We have fixed them in the revised PDF.

---

### Official Review · Reviewer_94ZC · 2024-10-30

**Soundness:** 3
**Presentation:** 3
**Contribution:** 2
**Rating:** 5
**Confidence:** 3

**Summary:**

The paper presents ScalingConcept which aims to enhance or suppress existing concepts in real input data using text-guided diffusion models. The proposed method leverages the concept removal and reconstruction capabilities of diffusion models to achieve concept scaling across both image and audio domains. The paper introduces the WeakConcept-10 dataset to evaluate the performance of the method and demonstrates its application in various zero-shot tasks such as canonical pose generation, object stitching, weather manipulation, and sound highlighting/removal.

**Strengths:**

- The paper conducts a thorough empirical analysis of the concept removal phenomenon in text-guided diffusion models, establishing a solid foundation for the proposed method.

- The ScalingConcept method is versatile, showcasing applications across multiple domains (image and audio) without additional fine-tuning.

- The introduction of the WeakConcept-10 dataset is a valuable contribution that provides a benchmark for evaluating concept scaling methods.

- The experiments are well-designed, with detailed comparisons against baseline methods and extensive ablation studies that illustrate the effectiveness of the proposed approach.

**Weaknesses:**

- While the method performs well on the WeakConcept-10 dataset, its generalizability to other datasets or more complex concepts remains uncertain. For example, in what scenario we will need this technique? It is applicable to combine removal and addition by existing methods to achieve the same performance. Further validation on diverse and challenging datasets would strengthen the paper.
In addition, the paper does not thoroughly address the scalability of the method to larger datasets or higher-resolution inputs, which could be a limitation for real-world applications.

- The complexity might limit its practical applicability. The reliance on precise hyperparameter tuning and inversion techniques may pose challenges for real-world applications.

- The method's effectiveness is highly dependent on the quality of text-to-X associations. In cases where the text prompt is not well-aligned with the model's understanding, the results might be suboptimal.

**Questions:**

- How sensitive is the method to the choice of hyperparameters (e.g., scaling factor and schedule)?

- The WeakConcept-10 dataset is a great start, but how does the method perform on more diverse and complex datasets? Have you tested it on standard benchmarks like COCO?

- Can you provide more insights into potential real-world applications of ScalingConcept? Especially the unique ones?

---

> ### Author Response · Authors · 2024-11-30
>
> ### More Diverse and Complex Datasets
> We thank the reviewer for their insightful question regarding the evaluation of our method on diverse and complex datasets. In the revised version of our paper, we have included additional experiments using the public and widely-used image editing dataset **TEdBench** (see Table 1, L412–419 in the paper). TEdBench contains diverse images across various categories and incorporates complex concepts.
>
> On this real-world dataset, we evaluate the performance of our method in concept scaling down, comparing it against existing methods (L442–462). The results demonstrate that our method generalizes effectively to other datasets and handles more complex concepts, validating the overall effectiveness of our approach. Moreover, our method surpasses existing methods in performance on this benchmark.
>
> ---
>
> ### Application Scenarios
> Our method provides two key functionalities (illustrated in Fig. 1):
>
> 1. **Concept Scaling in Input**:
>  Our approach achieves automatic editing suggestions, which are particularly useful when users want to explore potential editing effects related to the concepts in the input.
>
>
> 2. **Continuous Scaling Property**:
>  This property is valuable in scenarios where editing instructions are difficult to quantify using text alone. For instance, it allows adjustments in the presence (e.g., loudness) of an audio sound source within a mixture.
>
>
> ---
>
> ### Scalability
> We demonstrate the scalability of our method through experiments on both **concept scaling up** and **scaling down** across two datasets. Additionally, our **Application Zoo** showcases results in diverse domains, including but not restricted to:
> - **Natural Images**
> - **Face Images**
> - **Anime Images**
> - **Audio Spectrograms**
>
> Our method supports high resolutions, with image outputs of **1024×1024 pixels** and audio outputs up to **10 seconds** in length, reflecting its ability to handle complex data.
>
> ---
>
> ### Complexity and Sensitivity
> Our method does not require fine-grained hyperparameter tuning. Specifically:
> - For **scaling up** experiment in Table.1, we use $\omega_{\text{base}} = 5$.
> - For **scaling down** experiment in Table.1, we use $\omega_{\text{base}} = -5$.
> - Other hyperparameters, such as $\gamma = 3$ and $t_{\text{exit}} = 35$, are kept constant across experiments.
>
> These results show that a single set of hyperparameters is generalizable to different image types.
>
> Regarding inversion techniques, we use **ReNoise** as our default method. While not perfect, it delivers strong results, as evidenced by our experiments. Future advances in inversion techniques can be seamlessly integrated into our framework to further enhance performance.
>
> ---
>
> ### Dependence on Text-to-X Association Quality
> We acknowledge the reviewer’s concern about the dependence of text-guided diffusion models on the quality of text-to-X associations. The quality of text prompts is indeed critical, as suboptimal prompts may yield less-aligned results.
>
> This limitation is inherent to most editing methods relying on frozen, pretrained text-guided diffusion models, whose understanding is bounded by their training. Two potential avenues for improvement are:
>
> 1. **Automated Prompt Refinement**:
>  Leveraging tools such as LLMs to generate better prompts as a post-refinement step.
>
>
> 2. **Enhanced Text-Guided Models**:
>  Improving the language understanding capabilities of text-guided diffusion models, which aligns with the broader mission of advancing generative AI. Encouragingly, we are witnessing consistent performance improvements in this area each year.
>
> ---
>
> We hope this addresses the reviewer’s concerns. Thank you for your valuable feedback, which has helped strengthen our paper.

---

### Author Response · Authors · 2024-12-01

We thank all reviewers for their constructive and insightful feedback on our work. In response, we have uploaded a revised version of the paper with several key modifications and improvements:

1. We included a new experiment using the public TEdBench dataset to evaluate concept scaling down. Results comparing our method with baseline techniques are presented in **Table 1** and **Figure 6**.
2. We modified the introduction to better highlight the central research question of our paper: **How to scale inherent concepts in the input continuously.** This enables two key functionalities: Automatic Editing Suggestions and Continuous Scaling (as demonstrated in **Figure 1**).
3. We added two new applications in Application Zoo (Appendix A.1) to showcase the versatility of our method in diverse downstream tasks: Face Attribute Scaling and Anime Sketch Enhancement.

These additions are marked in blue throughout the revised manuscript (main paper and appendix).

In addition to these major updates, we also addressed minor issues, including typos, references, figure captions, and other points raised during the review process.

Once again, we extend our gratitude to the reviewers for their valuable comments and efforts in evaluating our work.

The Authors

---

### Meta-Review · Area_Chair_3t73 · 2024-12-21

**Metareview:**

- Scientific Claims and Findings:
    - This paper presents a new concept scaling method for text-to-image generation. The approach allows for the enhancement or suppression of concepts in input images by adjusting the noise difference between concept reconstruction and concept removal.

- Strengths:
   - The proposed method appears to be reasonable and effective.
   - The paper explores a novel aspect of scaling rather than replacing concepts, enhancing the application of diffusion models.
   - The introduction of the WeakConcept-10 dataset is a valuable contribution.

- Weaknesses:
    - The paper's writing and clarity could be enhanced. The motivation behind the work, particularly its connections to existing research, is somewhat unclear
   - The method needs to be validated across more diverse scenarios and datasets to demonstrate its robustness, applicability, and efficiency.

- Most Important Reasons for Decision:
     - Based on the identified weaknesses.

**Additional Comments On Reviewer Discussion:**

Although the rebuttal clarified the paper's motivation and provided additional experimental results and comparisons, all reviewers maintained their rating of 5.

Overall, the AC believes this paper has potential but would benefit from another round of major revisions to enhance its clarity and quality.

---

### Decision · Program_Chairs · 2025-01-22

Reject